

# Atmospheric Mixed Rossby Gravity Waves over Tropical Pacific
# during the Austral Summer
Hugo A. Braga[1] and Victor Magaña[1]
[1]Departamento de Geografía Física, Instituto de Geografía, Universidad Nacional Autónoma de México, 04510,  México City,
México.
*Correspondence to*: Hugo A. Braga  (hugoalvesbraga@icloud.com)
**Abstract.** Atmospheric Mixed Rossby-Gravity Wave (MRGW) activity during the austral summer months (Dec-Jan-Feb) is
examined by means of observational analyses for the 1991 - 2020 period. The main objective of the study is to explore the
relationship between tropical circulations at upper and lower tropospheric levels and tropical convective activity. Using an
Empirical Orthogonal Function (EOF) analysis of the high-frequency meridional component anomalies of the wind at 200 hPa,
for zonal wavenumber 5-6, episodes of intense MRGW activity are detected. Composite analyses based on an EOF analysis
show a quadrature phase over the central-eastern equatorial Pacific between the MRGW structure in the upper and lower
troposphere. Lagged correlations between the first two EOFs principal components, and the wind field and OLR, show that
MRGWs are laterally forced at upper tropospheric levels over the westerly duct region and later propagate westward and
downward. Once the MRGW reaches the lower tropospheric levels, it induces zones of moisture convergence that modulates
convective activity. Tropical convection develops in the divergent region of the MRGW at 200 hPa and in the MRGW moisture
convergence region at 700 hPa. Since the MRGW phase tilts eastward with height, moisture convergence at lower tropospheric
levels tends to coincide with divergence at upper levels favoring intense convective activity which results in the antisymmetric
outgoing longwave radiation anomalies off the equator near the MRGW. Therefore, the occurrence of MRGWs over the eastern
Pacific, is a form of tropical – extratropical interaction that generates tropical convection anomalies by means of induced lower
tropospheric moisture convergence and divergence anomalies.
**Keywords**: Mixed Rossby-Gravity Waves, Tropical–Extratropical Interactions, Lateral Forcing, Moisture Convergence.
## 1 Introduction
Equatorial waves are important elements of the atmospheric tropical circulations. Matsuno (1966) determined the main spatial
and temporal characteristics of Mixed Rossby-Gravity Waves (MRGWs), later identified by means of observational analyses
by Yanai and Maruyama (1966) and Maruyama (1967). MRGWs exhibit fluctuations in the meridional component of the wind,
with periods between 4 to 5 days and zonal wave numbers 4 to 6 (Yanai and Hayashi 1969; Yanai and Murakami 1970a, b;



Nitta 1970). Their vertical wavelengths range between 6 to 10 km (Holton 1979, Magaña and Yanai 1995) with an upward
propagation from the upper troposphere to lower stratospheric levels (Yanai and Hayashi, 1969).
The origin of MRGWs has been explored in several observational and model studies. Lateral forcing of MRGWs was originally
proposed in model studies by Mak (1969) and later explored by Bennet and Young (1971), Hayashi and Golder (1978) and
Zhang and Webster (1992), among others. Observationally, Yanai and Lu (1983), Magaña and Yanai (1995), Kiladis et al
(2009), Shreya and Suhas (2024) documented MGWs triggered by lateral forcing. On the other hand, tropical convective
heating has also been suggested as a mechanism that results in MRGWs (Holton, 1972; Hess et al., 1993). Hayashi (1970)
proposed that MRGWs could be the result of Wave-CISK, i.e., by means of the interaction between convective heating and
the wave itself. However, Takayabu and Nitta (1993) ruled out Wave-CISK as a mechanism to maintain MRGWs. Some of
these analyses also explore the role of MRGWs in modulating tropical convective activity (e.g., Magaña and Yanai 1995,
Kiladis et al 2009), but a definite answer to this problem has not been given.
The first observational studies on the vertical structure of MRGWs indicate that they extend from the troposphere to the lower
stratosphere (Yanai and Hayashi, 1969). A vertical node of these equatorial waves appears in the upper-tropospheric level
(around 200 hPa), and the phase tilts westward to lower tropospheric levels and eastward into the stratosphere (Magaña and
Yanai 1995; Zhou and Wang 2007; Kiladis et al., 2009). The tilting in MRGWs plays a crucial role in the vertical transport of
momentum and energy (Holton, 1979), but it may also be important in the spatial distribution of the convective activity
anomalies associated with MRGWs (Kiladis et al., 2009).
Recently, Shreya and Suhas (2024) confirmed that MRGWs are excited by lateral forcing from midlatitude disturbances from
the winter hemisphere. The propagation of these midlatitude waves into the tropics takes place across the eastern Pacific
westerly duct (Webster and Holton, 1982), which tends to remain "open" in the upper troposphere during the austral summer
months (Dec-Jan-Feb) (Braga et al., 2022). During the austral winter (Jun-Jul-Aug) the westerly duct periodically appears as
the Madden Julian Oscillation propagates along the eastern tropical Pacific, which allows the existence of MRGWs (Magaña
and Yanai, 1991). Therefore, lateral forcing as an excitation mechanism of MRGWs, takes place at upper tropospheric levels.
Eventually, MRGWs at higher tropospheric levels show in lower atmospheric levels, mainly towards the western Pacific region
(Au-Yeung and Tam 2018).
The processes involved in the downward phase propagation of a MRGW from upper to lower tropospheric levels, the near
boundary layer may modulate deep tropical convection. A study by Zhou and Wang (2007) shows that an upper tropospheric
MRGW acts as the precursor to a western Pacific tropical depression. Consequently, the downward phase propagation and the
vertical structure of the MRGW should be considered in the development of a region of intense convective activity around the
equatorial wave. These analyses suggest that as a MRGW, triggered at upper tropospheric levels propagates to lower
tropospheric levels, it may reflect in the modulation of moisture convergence and divergence near the boundary layer, that
ultimately controls deep convective activity in the equatorial regions.
As the westerly duct reaches maximum longitudinal extent and intensity, it allows the interhemispheric propagation of mid
latitude (Tomas and Webster, 1994; Li et al., 2015; Li et al., 2019; Braga et al., 2022) waves that are responsible for the




triggering of MRGW. Consequently, the existence of MRGWs and the corresponding antisymmetric anomalies in convective activity (Kiladis et al 2009) may be considered part of a tropical-extratropical interaction process.

The present study aims at examining the characteristics and evolution of MRGWs, in the Pacific region, and the relationship between this type of equatorial wave and convective activity off the equator, which remains as an open scientific question. This study is structured as follows: Section 2 outlines the characteristics of the data used for the study and the methodology of investigation. In Section 3, observational analyses are developed to determine the characteristics and evolution of MRGWs and their relationship with convective activity. In Section 4 summary and conclusions are presented.

## 2 Data

### 2.1 Data sets

For the identification of MRGWs, global reanalyses of daily tropospherics winds and specific humidity ERA-5 for the period 1991 to 2020 (Hersbach et al., 2020) have been used. The spatial resolution of ERA-5 wind data is $2.5° \times 2.5°$ from 1000 to 100 hPa. Daily Outgoing Longwave Radiation (OLR) data from the National Oceanic and Atmospheric Administration (NOAA) for the same period were also used (Liebmann and Smith, 1996) to document tropical convective activity anomalies.

### 2.2 Vertically Integrated Moisture Flux

The vertically integrated moisture flux field and its divergence were calculated to evaluate how atmospheric moisture is distributed by tropical disturbances in the tropical regions. The VIMF is a measure of the amount of water vapor transported in the atmosphere. Its convergence is used in the evaluation of the hydrological processes in the atmosphere (Fasullo and Webster, 2003). High VIMF convergence (VIMFc) zones are related to intense convective activity. The VIMF has been used to examine moisture transport processes, for instance in Easterly Waves (Pazos et al., 2023). The VIMF is calculated using the expression:

$$\text{VIMF} = \frac{1}{g} = \int_{p=1000}^{p=100} Vqdp \qquad (1)$$

where q is the specific humidity (kg·kg–1), V is the horizontal wind field, g is the gravity constant, and p is the pressure between 1000 and 100 hPa. VIMF units are $kg \cdot m^{-1} \cdot s^{-1}$.

### 2.3 Methodology

Various approaches have been used to diagnose MRGWs activity including spectral analysis with radiosonde data (Yanai and Hayashi, 1969), using reanalysis data (Yanai and Lu, 1983; Magaña and Yanai 1995; Wheeler and Kiladis, 1999), or by projecting meteorological wind fields of reanalysis data onto theoretical spatial structures of equatorial waves (Yang et al., 2003; Au-Yeung and Tam, 2018). MRGWs patterns may be identified by means of Empirical Orthogonal Function (EOF) analysis of the meridional component of the 200 hPa wind field. The Principal Components (PC1, PC2) of EOFs are used as



indices to compose wind, OLR, and atmospheric moisture fields to obtain the spatial characteristics of the MRGWs. The
identification of periods and regions of MRGW activity are determined based on the time series of intense signals in PC1 and
PC2. The temporal evolution of MRGW is examined by means of lagged – correlations of PC1 or PC2 and the wind, OLR and
moisture fields. For the present analyses, data are band-pass filtered with a Lanczos Filter (Duchon, 1979), in the period range
between 2 and 6 days. The spatial structure of the MRGW in the EOF analysis is captured with a spatial filter for zonal
wavenumbers 5 to 6 (Hayashi, 1982).

## 3. Results and Discussions

### 3.1 MGW Detection

The first EOF of the band-pass filtered component of the meridional wind (v) at 200 hPa, spatially filtered in the 5 and 6 zonal
wavenumber range, in the 10°N - 10°S, 80°E - 100°W domain shows the signal of a MRGW with a dominant zonal
wavenumber 5 (Fig 1). EOF2 also captures the MRGW signal but it is in quadrature with EOF1 which reflects the tendency
for the westward propagation of the wave. EOF1 explains 21.3% of the total variance, while EOF2 explains 19%.

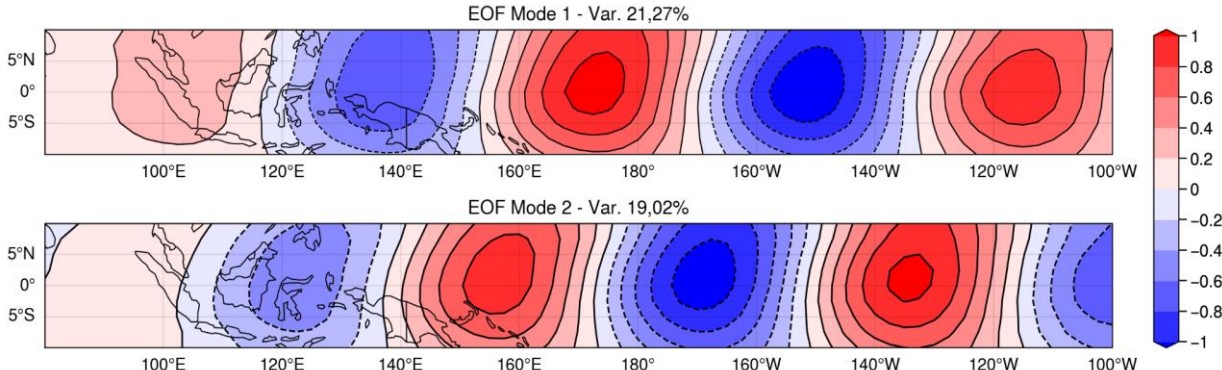


**Figure 1: First and second EOF for the 200 hPa space-time filtered anomaly of the meridional component of the wind field at 200 hPa for the December to February for the 1991 to 2020 period.**

To obtain the complete spatial structure of the wind field corresponding to a MRGW, composite patterns of the space-time
band-pass filtered wind fields at 200, 500 and 700 hPa were constructed using PC1 values above 1.0. The composite of the
wind field was combined with OLR anomalies at upper-levels and specific humidity anomalies at lower levels to obtain the
regions of the induced topical convective activity (Fig.2). Over the central eastern Pacific, a clockwise and an anticlockwise
circulation, centred along the equator show the expected anomalous wind field for a MRGW (Fig.2.a). In agreement with the
theoretical model of a MRGW, the convergent and divergent regions are anti-symmetrically located off the equator, between
the clockwise and anticlockwise circulations. The OLR anomalies associated with the MRGW coincide with the region of
convergence and divergence at upper tropospheric levels. At 500 hPa, the MRGW over the central Pacific is almost in phase
with its 200 hPa counterpart (Fig.2.b). When the signal of the MRGW is calculated at lower tropospheric levels it is observed
there is a phase shift towards the west in the clockwise and anticlockwise circulations. At 700 hPa, the intensity of the



circulations is weaker than at upper tropospheric levels, but the regions of atmospheric moisture convergence and divergence
are observed off the equator in between the cyclonic and anticyclonic circulations. Just above the tropical boundary layer, the
zone of moisture convergence (divergence) is located at 140°W and 5°N to 10°N (5°S to 10°S) and at around 100°W and 5°S
to 10°S (5°N to 10°N), on the west and east sides of the clockwise circulation in the central Pacific (Fig.2.c). Moisture
convergence-divergence leads to increases-decreases of atmospheric humidity that tend to coincide with the regions of
negative-positive OLR anomalies. Such connections between upper and lower tropospheric levels around the MRGW
circulation suggest that tropical convection anomalies are generated at lower tropospheric levels through moisture
convergence, that coincide with regions of divergence and convergence at upper tropospheric levels due to the quadrature in
the MRGW between these two tropospheric levels.

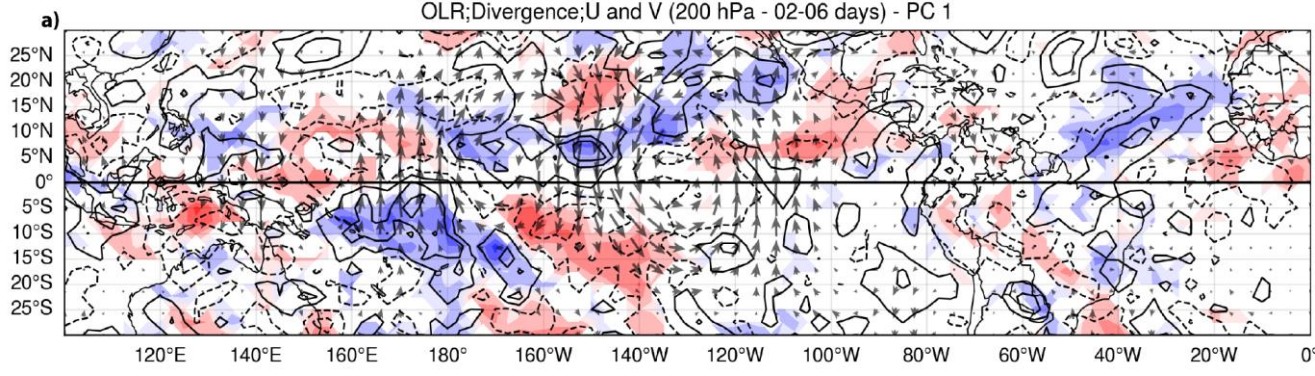

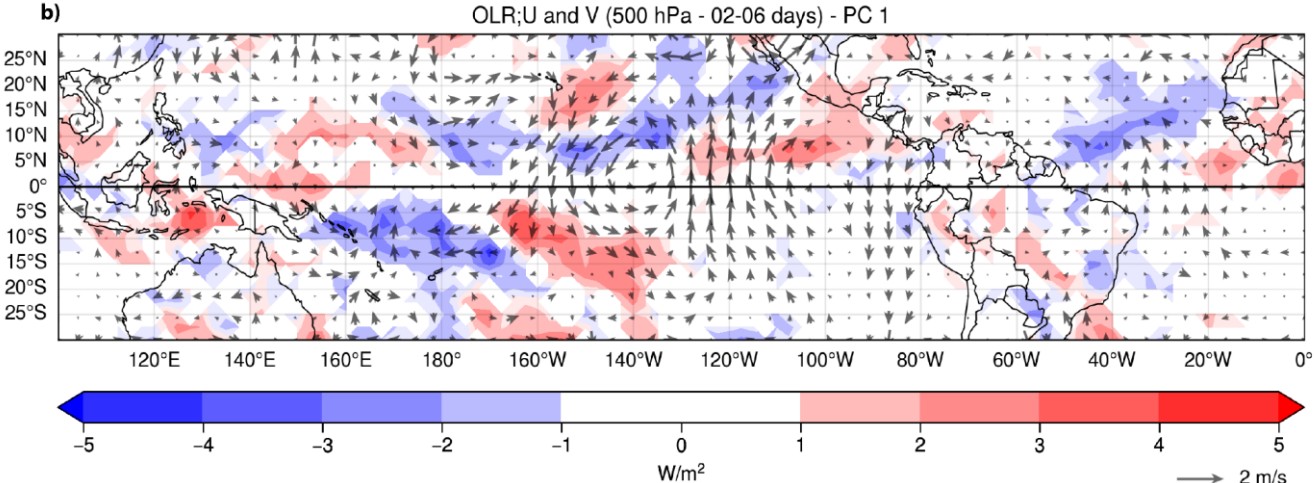






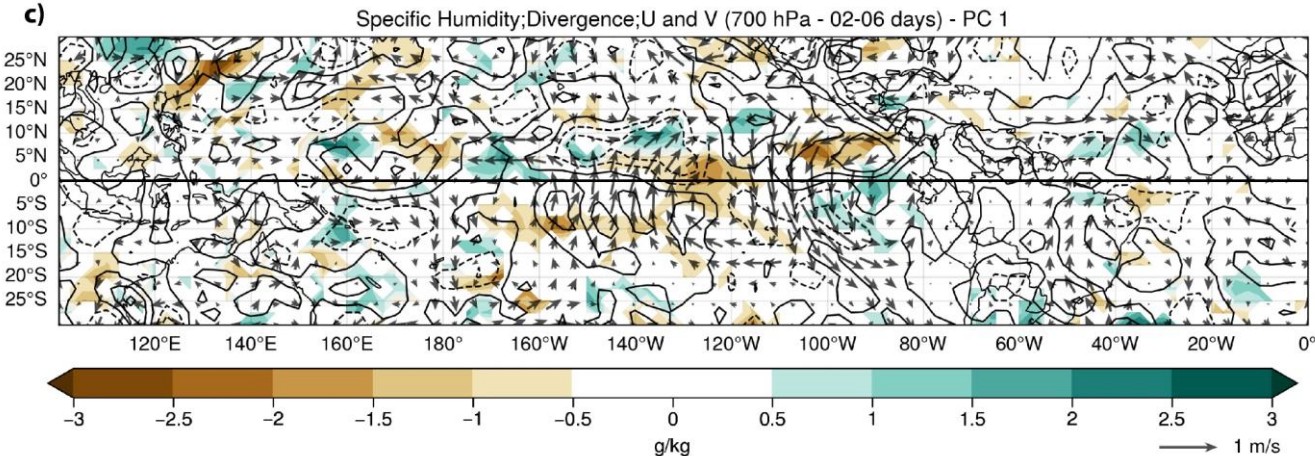

**Figure 2: Composite patterns based on PC1 > 1 conditions, showing band-pass filtered wind anomalies (2–6 day periods) at (a) 200 hPa, (b) 500 hPa and (c) 700 hPa. Wind anomalies are depicted as vectors, with dashed lines indicating convergence and solid lines showing divergence. Shading represents outgoing longwave radiation (OLR) anomalies (red and blue) and specific humidity anomalies (brown and green).**

As in previous analyses of the vertical structure of the MRGWs (Yanai and Hayashi, 1969; Zhou and Wang, 2007), the phase of the wave tilts eastward with height, from 700 hPa to 200 hPa, in the troposphere, while the phases tend to tilt westward from the upper troposphere to the lower stratosphere (Holton, 1979) reflecting the vertical structure of these equatorial waves. As MRGW evolves across the Pacific, the zones of convergence and divergence move westward along with the corresponding convective activity anomalies. The temporal evolution of a MRWG wind field and the associated tropical convection anomalies may be analyzed by examining the sequence of events, from the onset of the equatorial wave around 200 hPa, to a few days ahead, when its signal is observed at lower tropospheric levels. One point lagged correlations between PC2 and anomalies of 200 hPa wind and OLR in the -2 days to +2 days range show the evolution of a MRGW with an approximate 4 to 5 days period. At lag -2 days, the correlation with the wind field shows the MRGW pattern over the central eastern Pacific with vortices between 20ºN and 20ºS. In the central eastern an anticyclonic equatorial circulation is connected to a midlatitude wave that emanates from the northwest Pacific (Fig. 3a). The cyclonic of this midlatitude circulation mechanically couples with the MRGW that extends across the Pacific in a similar manner as the laterally forced MRGWs presented by Magaña and Yanai (1995); Kiladis et al. (2009) and Shreya and Suhas, (2024). Over the central Pacific, negative (positive) OLR correlations (anomalies) are observed between the clockwise and anticlockwise circulations of the MRGW over the equatorial region. The OLR anomalies are modulated by the midlatitude wave train, following the ascending and descending motions described by the omega equation. At lag 0 days, the MRWG propagates westward along with the antisymmetric anomalies in OLR around the dateline (Fig.3.b), with a westward phase velocity of approximately 10 m·s-1 and a zonal wavenumber 5. The modulation of OLR anomalies by the midlatitude wave that propagates across the eastern tropical appears to extend to the Southern Hemisphere to the western coast of South America, as previous analyses have shown (e.g., Braga et al 2022). At lag+2 days, the regions of convergence and divergence off the equator, around 15-20º in latitude, displace westward along with the tropical





152 convection anomalies (Fig. 3c). At this stage the midlatitude wave train weakens but the MRGW remains and is clearly

153 observed over the westerly duct region and the western Pacific.

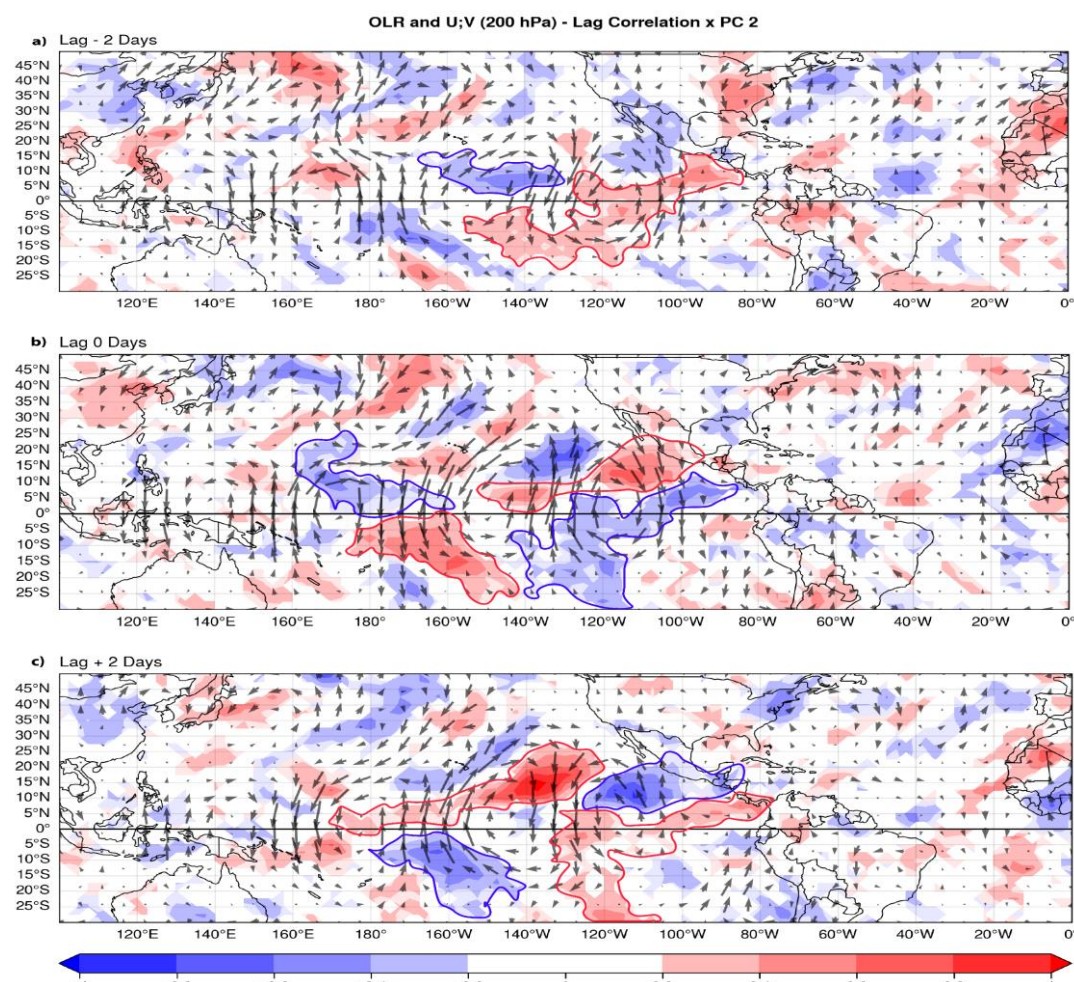

**Figure 3:** **Lagged-cross correlation between PC2 and the wind field at 200 hPa (vectors) and OLR anomalies (red and blue shading).**
**Contour interval: 0.2. The red and blue lines highlight the zone of antisymmetric OLR anomalies in the MRGW.**

The maximum amplitude of MRGWs at upper levels occurs over the westerly duct region at 200 hPa, but its amplitude
decreases over the Western Pacific, i.e., over a region with predominant easterly flow at upper tropospheric levels (Fig. 4). At
lower tropospheric levels, the westerly flow is observed over the western Pacific, where MRWGs have been documented
around 850hPa (Kiladis et al 2009). In this region, MRGWs may even lead to the formation of tropical cyclones (Dickinson
and Molinari, 2002; Zhou and Wang, 2007).






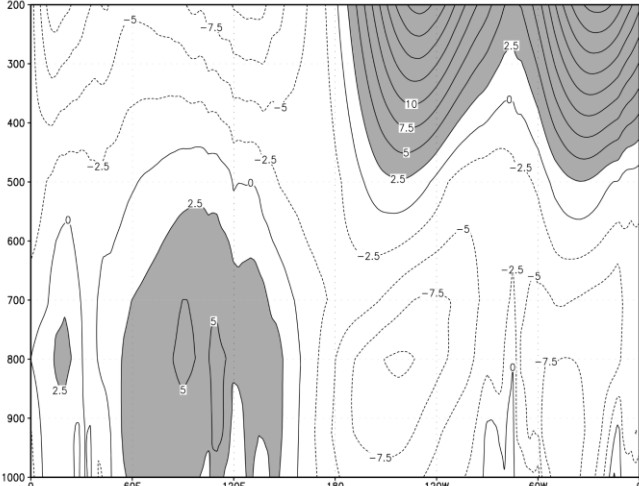

**Figure 4: Vertical cross section of the climatological zonal wind (m.s-1) along the equator between December and February 1991-2020. Shading corresponds to regions of westerlies.**

In the central eastern Pacific, MRGW activity is an important mechanism to modulate the lower tropospheric moisture field that results in tropical convection. Lagged cross correlations between PC1 and 700 hPa wind field and band-passed filtered anomalies of 700 hPa specific humidity show that in the central eastern Pacific MRGW modulate atmospheric moisture near the boundary layer. At lag -2 days, the sequence of clockwise and anticlockwise vortices corresponding to MRGWs along the equatorial Pacific begin to modulate the antisymmetric response in specific humidity over the eastern Pacific, around 100ºW (Fig.5.a). At lag 0 days, the correspondence between the antisymmetric structure of anomalous atmospheric moisture off the equator and the divergent and convergent regions associated with the MRGW is more evident. The MRGW extends from the central Pacific into the Atlantic and the antisymmetric atmospheric moisture anomalies are observed in the corresponding divergent and convergent regions off the equator, between 5º and 10º in latitude and around 100ºW. This equatorial disturbance exhibits a dominant zonal wavenumber 5 structure that extends into the Atlantic, in agreement with the westwards group velocity associated with MRGWs (Fig.5.b). At lag +2 days, the signal in specific humidity correlations moves westward, maintaining the antisymmetric structure off the equator and extending to the Atlantic Ocean (Fig. 5c). The antisymmetric structure in the specific humidity and divergence (around 140ºW) – convergence anomalies extends westward with the characteristic phase velocity of MRGW, while their signal extends into the tropical Atlantic and northern South America in relation to their group velocity.

**Figure 5: Lagged cross-correlation between the first principal component (PC1) of the 200 hPa meridional wind and the bandpass-filtered specific humidity (02–06 day periods; represented by shaded regions in green and brown), along with anomalies in the 700 hPa wind field (vectors), during the December–February period. Panels show results for: a) Lag -2 days, b) Lag 0 days, and c) Lag +2 days. Thick solid lines highlight regions of anomalous convective activity associated with the MRGW.**

## 3.2 Vertically Integrated Moisture Flux in the tropics

The modulation of moisture by low level MRGW circulations is also diagnosed by examining the Vertically Integrated Moisture Flux (VIMF) and its convergence. As previously stated, VIMF is a measure of the amount of water vapor transported in the atmosphere and its convergence is used to determine zones of intense convective activity. The lagged correlations of PC1 and VIMF and its convergence show that a MRGW tends to create regions of moisture accumulation that result in tropical



convective activity. By lag -2 days, the signals of a MRGW along the eastern equatorial Pacific and a midlatitude wave from
the northern subtropics along the westerly duct region show a tropical-midlatitude interaction (Fig.6.a). The positive and
negative vertical motion anomalies reflect in the regions of VIMF convergence and divergence in the midlatitude wave. In the
equatorial region, moisture convergence and divergence are located off the equator as expected in a MRGW. At lag 0 days,
VIMF and its convergence-divergence zones show the westward movement of the MRGW and the antisymmetric location of
the associated zones of moisture convergence and divergence (Fig. 6.b). The spatial structure of VIMF correlations
approximately match the one observed for the wind field anomalies at 700 hPa (see Fig.5.b) indicating that VIMF is capturing
the signal of MRGW at lower tropospheric levels. Such anomalous circulation modulates zones of specific humidity anomalies
off the equator in the central-eastern equatorial Pacific. By lag +2 days, the MRGW signal shows that moisture convergence
and divergence are asymmetrically distributed (Fig.6.c) contributing to increases and decreases of specific humidity, between
150°W and 70°W (see Fig. 5.c). The previous analysis shows that moisture is controlled by the MRGW at lower tropospheric
levels inducing zones of negative and positive convective activity anomalies. The quadrature between the phase of the MRGW
at upper and lower tropospheric levels serves to connect moisture convergence (divergence) at 700 hPa with divergence
(convergence) at 200 hPa, characteristics of deep tropical convective systems.

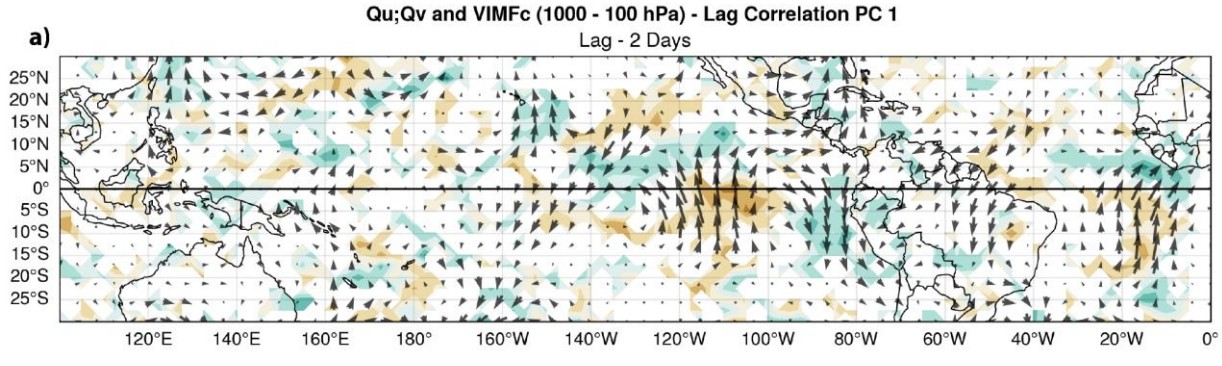


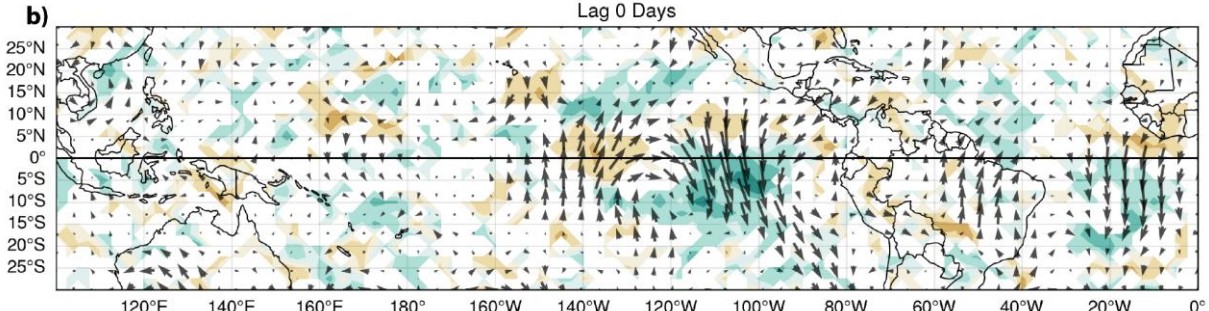




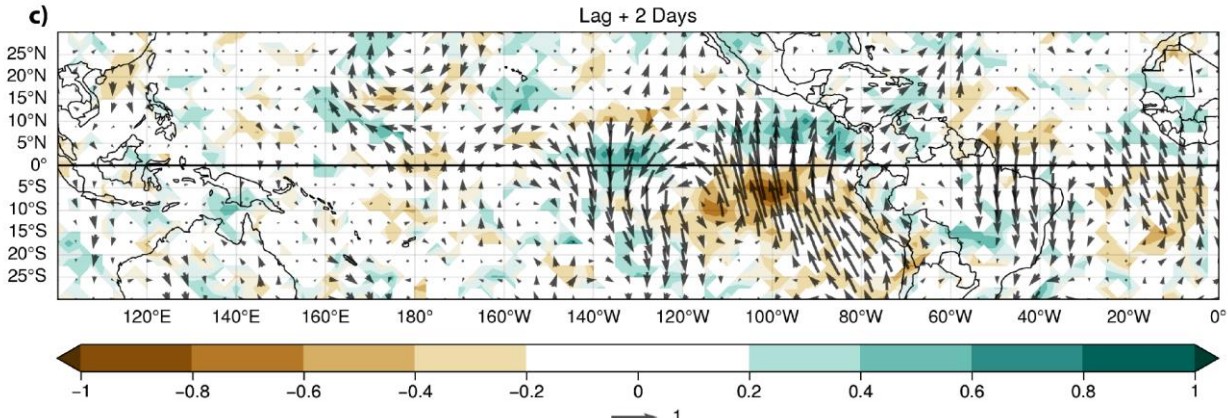

**Figure 6: Lagged-cross correlation between PC1 and VIMFc (vectors) and convergence (shades of green) and divergence (shades of brown) of VIMFc for the December – February months. a) Lag -2 days, b) Lag 0 days, and c) Lag +2 days.**

The relationship between VIMF convergence (VIMFc) and divergence, and OLR anomalies may be further examined by comparing the lag correlations between PC2 and VIMF convergence and OLR. For brevity, the relationship between VIMF and OLR correlations is shown only for lag +2 days. The signal of the midlatitude wave approaching the westerly duct region is observed as positive and negative correlations corresponding toVIMF convergence (divergence) and OLR negative (positive) anomalies. The signal extends into South America showing that not only it triggers a MRGW but also continues its interhemispheric propagation (Webster and Holton, 1982; Tomas and Webster, 1994; Li et al., 2015; Braga et al., 2022; Braga et al., 2024). Along the equatorial region the antisymmetric signals in correlation appear for VIMF convergence (divergence) and OLR anomalies extending from 150°W to 80°W.

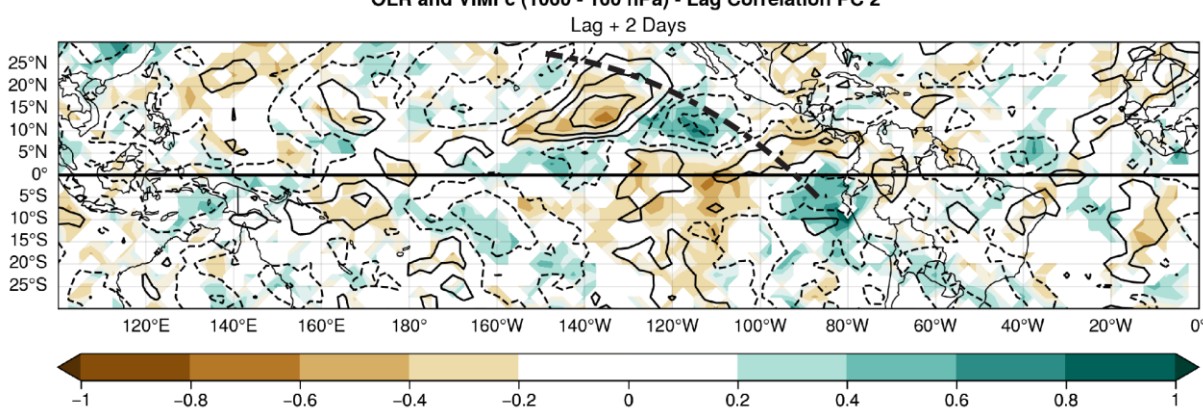

**Figure 7: Lagged correlation between the second principal component (PC2) and outgoing longwave radiation (OLR) at lag +2 days, represented by solid (positive) and dashed (negative) lines. Shaded regions depict vertically integrated moisture flux convergence (VIMFc).**



Over the eastern subtropical Pacific, in the region of tropical midlatitude interaction, positive signals of correlations for specific
humidity anomalies and OLR are observed extending into southern Mexico, that approximately correspond to the so-called
tropical plumes that at times correspond to precipitation events in this region (Knippertz, 2007; Fröhlich et al., 2013).

## 3.3 Case Study

An example of the presence of MRGWs in the daily atmospheric circulations is given in the wind field in upper and lower
tropospheric levels on February 2-6, 2020, identified as a large value of PC2. On February 2, 2020, a midlatitude wave over
the central-northeastern Pacific propagates into the tropics across the westerly duct region, coupled with the characteristic
circulation of a MRGW around 180ºW. At 200 hPa (180ºW-120ºW) OLR positive and negative anomalies are observed in the
regions of ascending and descending motions associated with the midlatitude wave that propagates from the northern to the
southern hemisphere (Fig.8.a). The clockwise equatorial circulation at 200 hPa corresponds to part of the midlatitude wave
but it is also a characteristic of the equatorial MRGW. At lower tropospheric levels (700 hPa), there is only a slight signal of
this clockwise circulation, almost in phase with the upper tropospheric vortex. At 700 hPa, the midlatitude wave is hardly
present in the wind field around the subtropics but it shows negative and positive specific humidity anomalies in the
convergence and divergence regions around 20ºN (Fig.8.b). By February 3, 2020, the midlatitude wave at 200 hPa extends to
the Pacific coast of South America with the corresponding positive and negative anomalies in OLR (Fig. 8c). At around 130ºW,
a well-defined vortex corresponds to an equatorial clockwise circulation with antisymmetric OLR anomalies. At 700 hPa, a
clockwise circulation may also be identified at 140ºW, with signals of a vortex that corresponds to the MRGW with positive
and negative specific humidity anomalies around (Fig.8.d). By February 4, 2020, the mid latitude wave in the Northern
Hemisphere subtropics remains, but it intensifies in the equatorial and the tropical Southern Hemisphere region (Fig.8.e). The
phase of the equatorial clockwise vortex around 130ºW appears to remain locked. In the lower troposphere the clockwise
circulation in the equatorial region is better defined and the anti-symmetric structure in the surrounding specific humidity
anomalies, characteristic of the MRGW, begins to form between 180ºW and 140ºW (Fig.8.f). On February 5, 2020, the MRGW
began its westward movement with anti-symmetric OLR anomalies better defined on its westward side. The midlatitude wave
signal in the north-central Pacific weakens (Fig.8.g). At lower levels the clockwise circulation associated with the MRGW is
well defined over the equator and shows a westward displacement with the specific humidity anomalies anti-symmetrically
distributed around this circulation, in the moisture convergent and divergent regions (Fig.8.h). The structure of the MRGW at
700 hPa appears to be better defined as it approaches the westerly winds, west of the dateline.

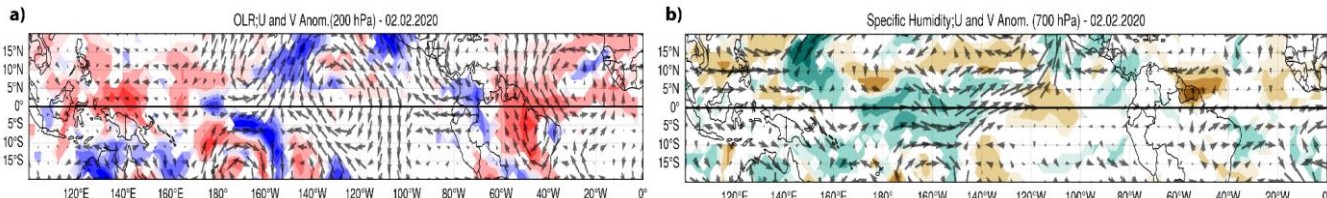



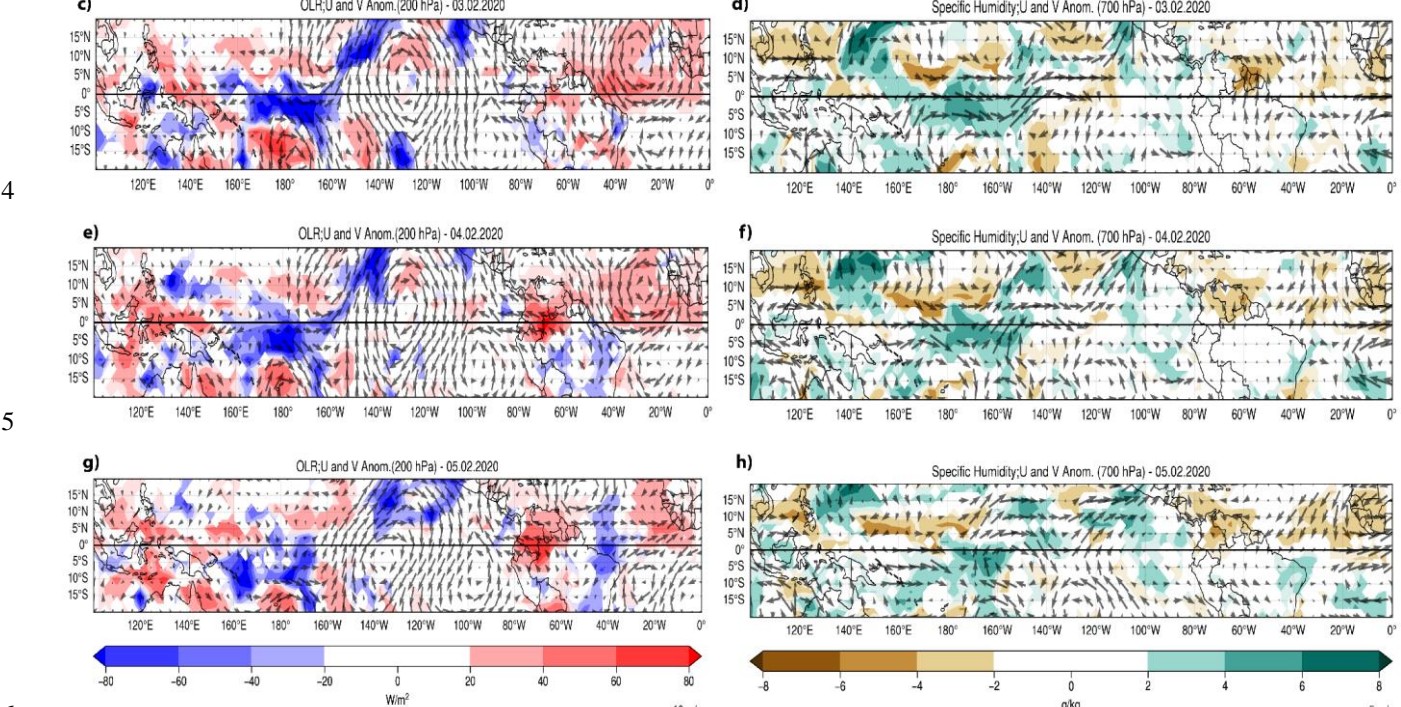

**Figure 8: Anomalies in atmospheric conditions (200 hPa wind and OLR in the left column, 700 hPa wind and specific humidity in the right column) from February 2 to 5, 2020: (a) and (b) for February 2, (c) and (d) for February 3, (e) and (f) for February 4, and (g) and (h) for February 5.**

As observed by Zhou and Wang (2007) in their case study of a MRGW, this equatorial wave is triggered at upper tropospheric levels and its signal propagates downward in the following days. The sequence of atmospheric circulations between 02-05 February 2020 shows that the equatorial wave circulations are well defined in the early days at 200 hPa, and its presence is better detected at later stages at 700 hPa. A vertical cross section of the meridional wind anomalies along the equator and specific humidity anomalies north and south of the equator (5ºN-5ºS) reflect the development of the vertical structure of the MRGW. The vertical cross section of the meridional wind anomalies between 5ºS and 5ºN (Fig.9.a) and specific humidity anomalies between 5ºS and 10ºS (Fig.9.b) and 5ºN and 10ºN (Fig.9.c), for February 2, 2020, shows that the signal of the MRGW in the wind field is present mainly at 200 hPa around 160ºW-100ºW, with magnitude of around 15 ms-1, between 100 and 400 hPa. At this stage of development, the specific humidity anomalies do not appear to correspond to the moisture convergence and divergence induced by a MRGW. By February 3, 2020, the signal of the MRGW in the central eastern Pacific, at upper tropospheric levels, extends downward to around 700 hPa (Fig.9.d). Some indications of the induced effect of the lower tropospheric part of the MRGW show in the specific humidity, with positive-negative anomalies around 700 hPa, between 160ºE and 140ºW (Fig.9.e). South of the equator the sign of these anomalies tends to be the opposite to its northward counterpart, but still is not well defined (Fig.9.f). By February 4, 2020, the quadrature in the anomalies of the meridional component of the wind field shows and extends to lower tropospheric levels with an eastward tilt with height at around 150ºW




(Fig.9.g). The tilt with height approximately corresponds to a vertical wavelength of around 15 km, which approximately
agrees with early estimates by Yanai and Hayashi (1969). At around 700 hPa positive and negative anomalies appear induced
by moisture convergence and divergence associated with the MRGW (Fig.9.h.i). On February 5, 2020, the structure of the
MRGW in the troposphere exhibits the tilt with height associated with the vertical wavelength between 180ºW and 120ºW
(Fig.9.j). North and South of the equator antisymmetric anomalies in the specific humidity field are well defined in association
with the circulations induced by the MRGW (Fig.9.k.l). This case study suggests that the moisture anomalies in the lower
tropospheric levels tend to develop as the MRGW propagates downward from the upper tropospheric levels where it was
triggered by a midlatitude wave. Once MRGW is well developed, it modulates moisture convergence and develops as deep
convection thanks to the wind divergence in the upper troposphere.





**Figure 9: Vertical cross-sections (longitude – height) (1000–100 hPa, between 5ºS–5ºN) for daily meridional wind anomalies (left column) and daily specific humidity anomalies (central and right columns) from February 2 to 5, 2020: (a), (b), and (c) for February 2; (d), (e), and (f) for February 3; (g), (h), and (i) for February 4; and (j), (k), and (l) for February 5.**

## 4. Summary and Conclusions

Upon the discovery of forced equatorial waves, numerous studies have proposed that they are forced either, by a midlatitude wave propagating into the tropics, or by convective activity near the equatorial regions. Lateral forcing appears to be a frequently accepted triggering mechanism for MRGW (Magaña and Yanai, 1995; Zhou and Wang, 2007; Shreya and Suhas, 2024). However, there is still some debate on the relationship between MRGW and the associated tropical convective activity. Even more, the signals of MRGW in the upper and lower troposphere are often treated separately.

The present studies show a plausible explanation to coherently relate all these various elements considering midlatitude wave forcing of equatorial circulation with the characteristics of a MRGW. As suggested by Au-Yeung and Tam (2018), the present studies shows that the signal of the MRGW extends to the lower troposphere where it changes the atmospheric moisture field, which in turn, results in antisymmetric anomalies in specific humidity off the equator, as those reported in other observational analyses (eg. Kiladis et al., 2009). The phase difference (quadrature) between the upper tropospheric MRGW circulations



(wind convergence-divergence) and its lower tropospheric counterpart (moisture divergence-convergence) reinforce the
development of deep tropical convection that shows as positive and negative OLR anomalies (Fig.10). Therefore, a key element
to associate convective activity and atmospheric circulation in a MRGW is the eastward tilt with height in the troposphere that
results in a quadrature of the phase of the wave. From the top of the troposphere to the stratosphere the westward tilt with
height corresponds to the vertical structure of the MRGW (e.g Holton, 1979).

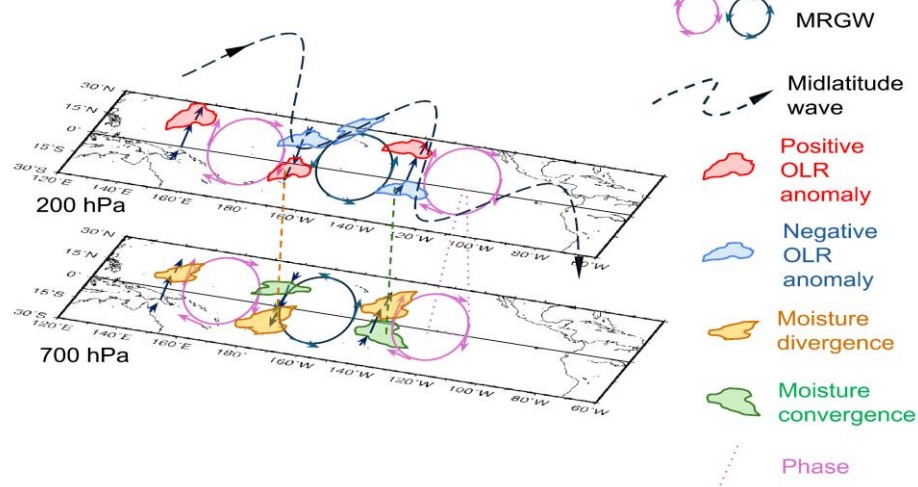


**Figure 10: Schematic of the vertical structure of a MRGW and the corresponding circulation anomalies in lower and upper**
**tropospheric levels along with moisture and convective activity signals in divergent and convergent regions of the wave.**

The development of MRGW constitutes a process that involves tropical midlatitude interactions that may be of relevance for
weather in the tropical region, where moisture convergence in the eastern tropical Pacific induced by MRGW constitute the
source of convective activity, even for tropical plumes observed over Mexico during the boreal winter season (Knippertz,
2007; Fröhlich et al., 2013). In addition, the propagation of the midlatitude wave into the Southern Hemisphere through the
westerly duct affects weather over South America (Braga et al., 2022).
Thanks to the improvement of atmospheric reanalysis, it is now possible to more accurately describe the characteristics of
equatorial waves in the troposphere and even in the stratosphere. A systematic identification of equatorial wave activity may
serve to better define the influence of these systems in weather in several tropical regions, for instance in the tropical Americas.
In summary, a key element of tropical weather in the eastern Pacific are MRGWs and consequently, a better understanding of
the processes of modulation of atmospheric moisture and convective activity may significantly improve weather forecasts in
the tropical and subtropical regions.

**Author Declaration**
**Funding information**: This work was supported by UNAM Postdoctoral Program (POSDOC), DGPA 13189. Victor Magaña.
was supported  by the CONAHCYT Grant PCC-319779



**Conflicts of interes**t:  There is no conflict of interest.
**Ethics approval**: All authors have approved this manuscript.
**Consent to participate**: All authors have provided their consent to submit this manuscript to Weather and Climate Dynamics.
**Consent for publication**: All authors give permission to publish this manuscript.
**Data availability**: Publicly available datasets were analyzed in this study. This data can be found here:
https://www.ecmwf.int/en/forecasts/dataset/ecmwf-reanalysis-v5.
**Authors' contributions**: Hugo A. Braga analzsed the data, wrote the manuscript, and prepared the figures. Victor Magaña
contributed some parts and reviewed the manuscript together with the first author.
**Acknowledgements:** We would like to express our gratitude to the Departamento de Geografia Fisica at the Instituto de
Geografia, UNAM for their support. The technical assistance provided by Gustavo Vázquez is highly appreciated. We also
thank DGAPA-UNAM 13189 for the postdoctoral fellowship that made this project possible, as well as CONAHCYT Grant
PCC-319779.

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
