# Peer review of "Atmospheric Mixed Rossby Gravity Waves over Tropical Pacific"

_EGUsphere, 2024_

## Author Comment (AC1)

Dear Reviewer #1 Thank you for your careful review of our manuscript. Your comments are greatly appreciated and we think this new version of the manuscript satisfactory responds to your concerns and provides an interesting contribution to the study of equatorial waves.

Responses to comments of reviewer #1 notes, are as follows:

1. The zonal spatial filtering to wavenumbers 5-6 is very restrictive.

**R: We have broadened the wavenumbers for the analyses to s= 4-6. We also did tests with zonal wavenumbers 3-8, and most of our results in the analyses remain. It means that the 4-6 zonal wavenumber signal is robust over the central eastern Pacific. Even more, we decided to apply EOFs to the 200 hPa meridional wind component with wavenumbers 4-6 without a bandpass filter. After that, we did a spectral analysis of PC1 and PC2 and results showed a spectral peak around 6 days, with high coherence square (0.6) between PC1 and PC2 in the (5 – 8 days)$^{-1}$ frequency range and PC2 leading by around 100$^{\circ}$ degrees PC1, i.e., they are almost in quadrature which corresponds to a westward phase propagation. The correlation between PC1 and PC2 acts as a means to filter the MRGWs signal in the correlations, for instance.**

[Figure]

Figure A1 - First and second EOF for the 200 hPa space-time filtered anomaly of the meridional component of the wind field at 200 hPa for the December to February for the 1991 to 2020 period.

Although the results appear to provide clear and robust connections to real signals over the Central Pacific Ocean, the Gibbs phenomenon would extended the filtered signal substantially east and west of the central point. In the real world, the MRG wave signals have a broad spectral footprint, with a peak in the power spectrum extending over planetary to small synoptic scales. Many previous works have shown substantial variance at the target band the authors are using. Yet individual disturbances have their zonal scales evolve across their lifetimes. For example, a disturbance moving westward across the Dateline near wavenumber 4 or 5 might arrive over the west Pacific, slowing down as it moves, ultimately projecting more strongly to narrower wavelengths, better characterized by wavenumbers 6 or 7.

**R: You are right. As we broadened the wavenumber domain in the analyses more spatial details appeared. Our case study shows that in the lower tropospheric levels the wavelength of their MRGW shortened and wavenumber 5-6 dominates over the western Pacific. We do not know the reason for this transition. However, over the central-eastern Pacific wavelengths corresponding to zonal wavenumber 4-5 dominate.**

**In the temporal domain,** the spectral peaks for PC1 and PC2 are around 6 days. Therefore, we decided to broaden the bandpass filtering to 3-8 days, applying a spectral filter to remove oscillations with periods above 90 days (Dec-Jan-Feb) and avoid the Gibbs Phenomenon. Results did not change significantly with respect to our previous analyses. Recurrently, we observed that at the 200 hPa level, the MRGW signal significantly weakened over western Pacific, in the easterly winds regime. There were no indications of a reduction in the wavelength of the MRGW signal at this level.

Figure 1 suggests that the authors' data are overfiltered, thus masking the scale change evident in previous works as the waves move westward.See Figure 12 of Kiladis et al. (2009) for an example. Although the central results of the authors over the middle of the Pacific basin conform well to previous works, Figure 1 does not allow for the disturbance to evolve in its zonal scale, because the wavenumber is over prescribed. It is unclear how this issue will impact the timing and other characteristics of the downward propagation of the disturbance that they diagnose.

**R:** **We have also made use of EOFs with zonal wavenumbers 3-8 in the 200 hPa meridional wind, and the corresponding PCs have been used to observe the spatial structure of MRGWs at lower tropospheric levels. The analyses do not show a change in the spatial structure over the central-eastern Pacific. It is only in the lower tropospheric levels that a transition to shorter wavelengths occurs as the MRGW propagates to the western Pacific (See figures A3 and A4). It is only in the case study that the dominant wavenumber at upper tropospheric levels is 6, while at lower tropospheric levels (700 hPa) the dominant zonal wavenumber close to the western Pacific is 7 (Fig 9g and Fig. 9j).**

**In all Lag Correlations no band-passed or spatial filters were applied to the wind or OLR fields in order to show the importance of the MRGWs in the winds and tropical convection fields. However, the correlations with PC1 or PC2 acts as a temporal and spatial filter of the signal.**

[Figure]

Figure A2 - First and second EOF for the 200 hPa space-time filtered anomaly of the meridional component of the wind field at 200 hPa for the December to February for the 1991 to 2020 period.

[Figure]

Figure A3 - Composite patterns based on PC1 > 1 conditions, showing band-pass filtered wind anomalies (2–6 day periods) at 700 hPa.

[Figure]

Figure A4 - Lagged cross-correlation between the first principal component (PC1) of the 200 hPa meridional wind along anomalies in the 700 hPa wind field (vectors) , during the December–February period.

The authors should broaden their wavenumber filter and repeat their analysis to assess the extent of the difference associated with the narrower scales that are evidently important as the disturbances move to the West Pacific. The filtering is likely not the only way the algorithm constrains results zonally. Even using a broader wavenumber filter, the EOF analysis will constrain the results to a particular range of zonal scales, but it will allow the zonal widths of the anomalies to vary geographically. Data filtered for a broad band along the MRG spectral peak ultimately expresses in several EOF pairs, each higher EOF pair

explaining progressively smaller zonal scales. This means that one pair of EOFs is not sufficient to describe the whole population of waves.

**R: The existence of a westerly duct during the Austral summer imprints special characteristics to the MRGWs. Specifically, the amplitude of these waves at upper tropospheric levels is larger over the westerly duct and the intensity of the MRGW band passed filtered wind anomalies is large as well, of the order of 2 ms$^{-1}$ at 200 hPa and 1 ms$^{-1}$ at 700 hPa. However, when unfiltered anomalies are used, the signal of the MRGW is present, with wind anomalies of the order of 10 ms$^{-1}$ and 5 ms$^{-1}$ at 700 hpa. Total wind associated with the MRGW for the case study at 200 hPa is around 50 ms$^{-1}$, and 10 ms$^{-1}$ (See Figure A5). Therefore, the signals of MRGWs obtained in the analyses are not the result of over-filtering. In any event, you are correct, the signal in the filtered and unfiltered wind field associated with MRGWs at lower tropospheric levels propagating into the western Pacific show a smaller wavelength that over the eastern Pacific at 200 hPa. Its is not the objective of this study to explore shorter wavelengths that are not necessarily part of the MRGWs (For instance when they tends to result in tropical cyclones). It is interesting though, to specifically study the transition to shorter wavelengths particularly at lower tropospheric levels over the western Pacific. With a broader spectrum of zonal wavenumbers there may be smaller scale details in the analyses, in higher EOFs, but they may not necessarily have a simple physical meaning, and may be just an result of the orthogonality of the method.**

[Figure]

Figure A5 - Case Study total wind vectors 04.02.2020.

There is nothing wrong with the authors emphasizing a particular range of these scales through selecting a single pair of EOFs, but they should acknowledge that MRG energy also occurs at longer and narrower wavelengths than those that they show here.

**R: We have added a brief sentence on the transition of the wavelength at lower tropospheric levels as an interesting problem to be explored. L.298-303**

EOFs based on data filtered for a broader band of wavenumbers will still have leading modes concentrate at wavenumbers 4, 5, or 6 over the central Pacific, but the individual modes will associate with signals at narrower scales as the disturbances move westward.

**R: We are now using zonal wavenumbers 4 to 6 for the EOF analysis.**

2. The MRG wave exhibits eastward group velocity, not westward. The manuscript appears to state that the group velocity is westward.

**R: We appreciate the correction and have changed to "Eastward" on L.177 (Now L182).**

The wavenumber filtering and the EOF analysis selects for wave scale in a particularly narrow way, which will mask the development of the group velocity in their results. If the authors filtered for a broader wavenumber band, a pair of EOFs would still select for a particular narrow range of wavenumbers (even though the patterns would allow the same disturbance to be characterized by different wavelengths in different regions). In that case, analysis of multiple pairs of EOFs of MRG filtered data retained together would reveal the group velocity as the interference pattern that emerges from including wave signals propagating at different phase speeds over a range of zonal wavenumbers.

**R: The broadening of zonal wavenumbers to s = 4 – 6 allows a more adequate description of the group velocity from the eastern equatorial Pacific towards the Atlantic ocean.**

---

## Author Comment (AC2)

Dear Reviewer #2 Thank you for taking the time to carefully review our manuscript and for your insightful comments. We appreciate your positive feedback on the study's contributions and your constructive suggestions. Below, we address your main comments and detail the revisions made. We consider this improved version of the manuscript is clearer and adequately responds to your concerns and sends an interesting scientific message.

Responses to comments of reviewer #2 notes, are as follows:

This is an interesting study of the statistical structure and behavior of tropospheric mixed Rossby-gravity (MRG) waves over the eastern equatorial Pacific during northern winter. Using EOF analysis, the authors have obtained some nice results related to the extratropical forcing of MRG waves within the westerly duct. However, I think that the zonal wavenumber 5-6 filtering they used is unnecessarily narrow, and this has the potential to distort the actual scales of MRG waves compared to what has been shown in past studies (see comments below). In addition, the descriptions of the methodology used are incomplete, and while I was finally able to back out what they are actually using in their approach, this should be made more obvious to the reader at the outset in Section 3.1. Furthermore, a lot of relevant literature has not been cited, and I think the authors need to compare their results to those from these past studies. I recommend revisions of this manuscript while taking into account the comments by line number below:

25: The term "WMRG" has also been used to identify westward propagating MRGs by Yang et al. starting in 2003, to distinguish from the eastward propagating EIGs of Matsuno's n=0 meridional mode. MRG-E has also been used to describe the eastward propagating side by Knippertz et al. (2022), for example. If you are going to use MRGW instead of simply MRG, then I think it would be important to make this point in order to avoid confusion.
**R: We appreciate the suggestion and have updated it to MRGWs on L.25. (Now L.26-27).**

29: I think it's important to distinguish convectively coupled MRGs in the troposphere from the free MRGs in the stratosphere. The early studies focused on MRGs in the stratosphere, and these have decidedly different scales from those in the troposphere (e.g., Wheeler et al. 2000, their Fig. 12; Yang references below; Kiladis et al. 2016).
**R: We have updated it on L.25. (Now L.26-27).**

33: MGWs => MRGWs.
**R: We updated it to MRGWs on L.33. (Now L.36).**

33: Kiladis et al. 2009 did not document lateral forcing but mentioned previous studies that did. More recent examples include Yang and Hoskins (2016), Yang et al. (2018), Kiladis et al. (2016), and Suhas et al. (2020). Suggest citing these here for completeness, as these will also become relevant below.
**R: We appreciate the suggestion and have updated it on L.34-35.**

89: Examples of different methods employed are discussed in detail in Knippertz et al. (2022).
**R: We have updated it on L.89.**

90: As was done in Kiladis et al. 2016.
**R: We have updated it on L.90.**

Based on what is stated on line 107, it appears that you are using a correlation matrix and not a covariance matrix for the EOF analysis. This should be stated here.
**R: Thank you for pointing this out. We used a covariance matrix, and we have now included a corresponding statement in the methodology section.**

91: Not evident in this statement is the important point that EOF analysis of propagating disturbances will generally yield two EOFs in spatial and temporal quadrature, which is why you can use the combined PC1 and PC2 as an activity index.
**R: We have now used EOFs of meridional wind component with zonal wavenumber 4-6 bandpass filtered in the 03-08 days$^{-1}$ . We also calculated EOFs without bandpass filters and applied spectral analyses to the PCs. Results indicate that PC2 leads PC1 by around 100 degrees which implies quadrature between EOF1 and EOF2, and even more, that there is a signal of westward phase velocity.**

95: What is the basis for the spatial and temporal filtering? 2–6 days can be justified in the troposphere, but using only zonal wavenumbers 5 and 6 is very restrictive. In 1982, Hayashi had limited knowledge of the spatial scales of MRGs, which we now know are localized wave packets comprised of a number of zonal wavenumbers in both the troposphere and stratosphere. While spectra of meridional wind at 200 hPa do have power concentrated on wavenumber 5, the power is broad-band and extends especially to lower wavenumbers (Randel 1992). Indices based on antisymmetric OLR (Kiladis et al. 2009, 2016) or dynamically based indices (Yang et al. 2003; Knippertz et al. 2022) generally include wavenumber 1–4 components as well, and the structures of MRGs obtained in these studies are generally broader in scale than what is obtained here. I think you need to reconsider your filtering by including a broader zonal wavenumber range while testing the sensitivity of your results to these choices. It seems to me that including wavenumbers 1–4 initially would be a good test. In the 2–6 day period range, it is probably not necessary to only include westward propagating wavenumbers, but I suggest testing that approach as well. In other seasons, such as northern summer, this broader band filter would also include tropical depression (TD-type) disturbances that MRGs often morph into, but that should not be an issue during DJF.

**R: We have broadened the wavenumbers for the analyses to s= 4-6 and the temporal filtering to 3-8 days$^{-1}$ . We also tested with spatial wavenumber 1-4 and temporal filtering 2-6 days$^{-1}$ (See Figure A1). However this spatial pattern does not reflect the wind structure for MRGWs events. This spatial temporal domain acts as a filter to emphasize the signal of MRGWs only. Its is clear that other forms of variability in space and time are part of the wind field and tropical convection field. But the main objective is still to connect upper and lower tropospheric signals of MRGWs and convection only.**

**Yes, the development of MRGWs into TDs is an interesting problem and our results (case study) appears to indicate that a transition to shorter wavelengths over the western Pacific occurs at lower tropospheric levels. However this would require further analyses on how such transition takes place.**

[Figure]

Figura A1 - First and second EOF for the 200 hPa space-time filtered anomaly of the meridional component of the wind field at 200 hPa for the December to February for the 1991 to 2020 period.

98:"MRGW"

**R: We have updated it on L.102.**

99:Was the meridional wind area-weighted by the cosine of latitude? While this may not make a difference at the latitudes used, it is still a standard practice that should be followed. Additionally, justification for the chosen domain should be provided.
**R: Yes the cosine factor is included in the calculation of the divergence. The domain was selected based on previous studies like Killadis et al. 2016, Suhas et al. 2020. We have updated it on L.103-104.**

101: It should be noted that PC1 would lead PC2 in the case of a westward-propagating disturbance.

**R:In the updated EOF analysis PC2 leads PC1 by around 100 degrees (quadrature). This was the result of a spectral analysis between PC1 and PC2. L.106**

102: What is the lagged correlation between PC1 and PC2? While it is likely quite high, providing this value would further justify using the combined first two EOFs as an index.

**R:The coherence squared between PC1 and PC2 around the 6 days period is almost 0.8 and between the 5 to 8 days period is 0.66. L.107.**

107: Consider pointing out that the standard deviation of PC1 will equal one when using a correlation matrix (or standardized input). I assume you are compositing based on local temporal maxima in PC1?

**R: Yes, the standard deviation of PC1 is almost 1 and that is why we choose events with PC > 1.0 to compose the MRGW patterns.**

The figure caption mentions that the winds are bandpass-filtered, but is this also true for humidity and OLR? More details on the compositing technique are needed here.

**R: Yes, specific humidity and OLR were band-passed filtered. It is now stated so in the figure caption.**

110: The patterns compare favorably with those obtained by Wheeler et al. (2000) and Kiladis et al. (2009, 2016) using OLR or brightness temperature as the basis, including the diagonal tilt of the OLR signals. However, the circulation gyres appear to be smaller in scale, likely due to the wavenumber filtering applied here.

**R: With the broadened wavenumber filter, and now, the vortex appears to be slightly larger than when we used the previous filter. Still zonal wavenumber 5 dominates the pattern.**

124: I do not clearly see the quadrature relationship referenced in the text in Fig. 3. It seems more like divergence is out of phase vertically without much longitudinal displacement between moist and dry regions, as also reflected in the locations of the OLR anomalies.

**R: We suppose you refer to figure 2, where we compare the composite pattern at various vertical levels. The phase difference between 200 hPa and 700 hPa is around 20 degrees. Considering the wavelength of wavenumber 5 MRGW is around 75 degrees, the upper and lower signals of the MRGWs are in quadrature.**

138: From the scale, it is clear that these are lag correlations (not lagged regressions) for OLR.

**R: You are correct, we calculated lag correlations.**

However, how is the wind field being scaled using correlations? Does a vector length of "1" represent a perfect correlation for both uuu and vvv? Much more detail is needed here.

**R: Thanks for your comment, we have added further information to explain the characteristics of figure 3. L.142-145. and in all lag correlation figure captions.**

Additionally, are the "anomalies" bandpass-filtered to 2–6 days?

**R: The anomalies are not filtered in the lag correlations.**

144: Kiladis et al. (2009) does not show midlatitude coupling with MRGs, but Kiladis et al. (2016) does.

**R: We have changed it in L.144. (Now L.150) .**

147: In what sense does the omega equation hold? It appears that negative OLR occurs ahead of troughs, as expected.

**R: This is correct, We have updated it on L.147. (Now L.154).**

148: A phase speed of 10 m/s is significantly slower than the 15–25 m/s reported in previous studies. This may be due to the restriction to wavenumbers 5–6, which would inherently yield a slower phase speed based on MRG dynamics.

**R: With the broadened zonal wavenumber filter, the estimated phase speed of the MRGW is around 15 m/s. We have updated it on L.148. (Now L.156).**

150: Are you referring to the weak OLR anomalies off the west coast of South America in Fig. 3c? This does not appear to be a particularly strong signal.

**R: We have deleted the reference about OLR anomalies over South America.**

176: "Eastward group velocity"

**R: We have updated it to "eastward" on L.177 (Now L182).**

181: I believe you mean that the group velocity causes an MRG to form over the Atlantic, which then exhibits the characteristic antisymmetric specific humidity field associated with it.

**R: You are right. The signal of the MRGW extends to the equatorial Atlantic due to the eastward group velocity. L.184-185.**

215: You should reference Fig. 7 here. I do not understand why PC2 is being used for Fig. 7, as it cannot be directly compared with the circulation shown in Fig. 6c, for instance. Is there a justification for this?

**R: We have changed this analysis to PC1.**

219: A comparable statistical 200 hPa sequence for MRG activity farther west during DJF is also shown in Kiladis et al. (2016), their Fig. 16.

**R: We have changed it in L.219. (Now L.225)**

226: I do not see the humidity and OLR signals over southern Mexico that you refer to—are you referencing Fig. 6 or Fig. 7?

**R: We changed to a correlation between PC1 and VIMFc and OLR, Figure 7.**

234: Once again, are these anomalies bandpass-filtered to 2–6 days? This should be explicitly stated.

**R: The anomalies are not filtered in the case study, we state so in the figure caption.**

244: Do you mean it's a standing wave?

**R: The center of the clockwise circulation remains at the same longitude for only 2 days, so it is difficult to refer to it as a standing wave. To avoid confusion we have deleted that statement.**

305: Please refer to Yang and Hoskins (2017) for a discussion of the eastward tilt in the height of MRGs within the westerly duct during December–February.

**R: We have changed the statement in L.305. (Now L.320).**

**References**

- Kiladis, G. N., J. Dias, and M. Gehne, 2016: The relationship between equatorial mixed Rossby-gravity and eastward inertio-gravity waves: Part I. *J. Atmos. Sci.*, 73, 2123–2145.
- Knippertz, P., M. Gehne, G. N. Kiladis, K. Kikuchi, A. R. Satheesh, P. E. Roundy, G.-Y. Yang, N. Žagar, J. Dias, A. H. Fink, J. Methven, A. Schlueter, F. Sielmann, and M. C. Wheeler, 2022: The intricacies of identifying equatorial waves. *Quart. J. Roy. Met. Soc.*, 148, 2814–2852.
- Randel, W. J., 1992: Upper tropospheric equatorial waves in ECMWF reanalysis. *Quart. J. Roy. Met. Soc.*, 118, 365–394.
- Suhas, E., J. M. Neena, and X. Jiang, 2020: Exploring the factors influencing the strength and variability of convectively coupled mixed Rossby–Gravity waves. *J. Climate*, 33, 9705–9719.
- Wheeler, M., G. N. Kiladis, and P. J. Webster, 2000: Large-scale dynamical fields associated with convectively coupled equatorial waves. *J. Atmos. Sci.*, 57, 613–640.
- Yang, G., and B. J. Hoskins, 2016: ENSO-related variation of equatorial MRG and Rossby waves and forcing from higher latitudes. *Quart. J. Roy. Met. Soc.*, 142, 2488–2504.
- Yang, G., and B. J. Hoskins, 2017: The equivalent barotropic structure of waves in the tropical atmosphere in the Western Hemisphere. *J. Atmos. Sci.*, 74, 1689–1704.
- Yang, G., J. Methven, S. Woolnough, K. Hodges, and B. J. Hoskins, 2018: Linking African Easterly Wave activity with equatorial waves and the influence of Rossby waves from the Southern Hemisphere. *J. Atmos. Sci.*, 75, 1783–1809.